# The Effects of Inter-Set Recovery Time on Explosive Power, Electromyography Activity, and Tissue Oxygenation during Plyometric Training

**DOI:** 10.3390/s21093015

**Published:** 2021-04-25

**Authors:** Shuo Guan, Nan Lin, Yue Yin, Haibin Liu, Liqing Liu, Liping Qi

**Affiliations:** 1School of Biomedical Engineering, Dalian University of Technology, Dalian 116024, China; guannim@dlut.edu.cn (S.G.); musicyy1997@mail.dlut.edu.cn (Y.Y.); 2Faculty of Physical Education, Dalian University of Technology, Dalian 116024, China; ln920928@yeah.net (N.L.); liuhaibin@dlut.edu.cn (H.L.)

**Keywords:** wavelet analysis, near-infrared spectroscopy, plyometric, tissue saturation index, jump height

## Abstract

Performing continuous sets to failure is fatiguing during the plyometric training. Cluster sets have been used to redistribute total rest time to create short frequent sets so that muscle fatigue can be avoided. The purpose of the study was to investigate the effects of inter-set recovery time on lower extremity explosive power, neuromuscular activity, and tissue oxygenation during plyometric exercise and recovery. An integrated assessment of explosive power, muscle electrical activity, and tissue oxygenation was adopted in the present study to help understand local muscle metabolism and fatigue during plyometric exercise and recovery. Ten university male basketball players participated in this study. Subjects performed 4 groups of exercise, each group comprised of 3 sets of jumps: 1, 2, 3, or 5 min. Surface electromyography (sEMG) signals were collected from 9 lower extremity muscles; near-infrared spectroscopy (NIRS) was recorded on *vastus lateralis*; mechanical data during plyometric exercise were collected from a force plate. No significant differences among sets and among groups were found regarding explosive power, jump height, EMG intensity, mean power frequency, the rate of tissue saturation index, and HbO_2_ changes between baseline and recovery. The current study has shown no muscular fatigue induced during the 4 groups of exercise. The results of this study may help inform recommendations concerning the recovery time during plyometric exercises at low loads (30% 1 RM).

## 1. Introduction

Plyometric training is a widely used method to develop explosive strength using natural dynamic movements such as jumping and countermovement jumping (CMJ) [1]. Two training methods, strength, and plyometric training are usually recommended in literature to achieve the most powerful explosive strength [2]. Strength and plyometric training methods aim to help athletes to jump higher and/or further, to run faster, throw further and lift heavier weights. Several studies have indicated that regular participation in strength training and/or plyometric training programs can improve jumping and sprinting performance and also increase strength, independent of the age and playing level of basketball players [3,4,5]. In addition, it has been shown that a combination of weight and plyometric training constitutes a more effective method for enhancing basketball players’ motor skills than stand-alone approaches [6,7]. Previous studies have called for combined weight and plyometric training as an alternative to biomechanically comparable low-load strength training exercises with plyometrics in the same set [8]. Many coaches and athletes make use of combined weight and plyometric training to enhance power and speed [4,9,10,11]. The combined weight and plyometric training can also be used to improve agility and coordination for better athletic performance [12,13]. Well-designed and sport-specific plyometric training can be used as a safe and effective training method for improving jumping, sprinting, and agility performance in team sports athletes. For the present study, we have chosen loaded counter-movement jumps (CMJ) as a daily training method. The CMJ refers to a vertical jump in which the individual initially moves down, counter to the direction of the jump, upon reaching a determined angle the individual then forcefully and without hesitation jumps upward [14]. Loaded counter-movement jumps combine strength and plyometric training, where added loads can enhance an individual’s lower body muscle coordination and the ability to rapidly increase muscle tension for a greater maximal rate of force development [4]. Loaded CMJ jump can be considered one of the most ‘‘explosive’’ tests due to both its very short duration and the high intensity involved. Several studies have examined the training volume in terms of the number of sets, frequency, and intensity of plyometric training undertaken to achieve greater power [15,16,17,18]. It has been shown that the peak power is achieved with high loads of 70–90% 1 repetition maximum (1 RM) during weight training [19,20]. However, the peak power occurs at relatively lighter loads, between 0 and 60% 1 RM back squat [21,22,23,24,25]. In terms of training volume, weight training with lower loads (<60 1 RM), lower volumes (2–3 sets and 4–6 repetitions per set), and higher explosive velocity induce similar strength gains and greater improvements in jump and sprint performance compared with high load weight training [7,26,27]. However, little research has been done concerning the effects of inter-set recovery time on athletes’ performance during plyometric exercises. The inter-set recovery time is important to avoid muscle fatigue. Muscle fatigue could be beneficial in promoting muscle growth (such as bodybuilding) but it is usually harmful, causing serious injury when the level of fatigue is high [28]. Some researchers have suggested that inter-set recovery time during plyometric training should be two to four minutes to avoid muscle fatigue that may lead to incorrect technique, unnecessary risk of injury, over-reaching, and over-training [1,29].

During the training exercise, the changes in surface electromyographic (sEMG) signals can reflect the state of muscle activity in real-time [30]. The assessment of muscle fatigue based on the sEMG signals has been well reported in comprehensive reviews [31,32]. Two EMG parameters, mean power frequency (MPF) and amplitude, have been studied extensively as a means of tracking the progression of exercise-induced muscle fatigue over time [33,34,35]. sEMG studies investigating muscle activation show substantial increments during fatiguing activity. The increase of sEMG intensity is associated with muscle fatigue due to increased motor unit recruitment, increased motor unit firing rate, increased motor unit synchronization as well as the increased low-frequency content which more readily passes through the low-pass spatial filter of the muscle and surrounding tissue [36]. The decrease of sEMG MPF was related to fatigue due to changes in intramuscular pH, modification in the recruitment and synchronization of motor units, and fiber type [37].

In addition, the ability to maintain a high intensity throughout multiple exercise sets depends on the recovery from one exercise set to the next. Oxygen plays a vital role in the musculoskeletal system during exercise and recovery, and muscle function can be critically affected if oxygen supply is limited [38]. Near-infrared spectroscopy (NIRS) is a non-invasive optical measurement method of blood oxygenation based on changes in the near-infrared absorption characteristics of oxygenated hemoglobin (HbO_2_) and deoxygenated hemoglobin (HHb) [39]. At 750 nm, deoxygenated hemoglobin (HHb) has a much higher light absorbance while at 850 nm oxygenated hemoglobin (HbO_2_) has a higher absorbency [40]. Measurement of HbO_2_ and HHb by NIRS has been used to non-invasively monitor specific regional tissue oxygenation changes during exercise and recovery [40,41,42].

The commonly recommended inter-set recovery period for plyometric training is between 2 min and 4 min [23,25,43,44]. The purpose of this study was to investigate the effects of inter-set recovery time on lower extremity explosive power, neuromuscular activity, and tissue oxygenation during plyometric exercise and recovery. An integrated data collection and analysis of explosive power, electromyography activity, and tissue oxygenation is necessary for understanding local muscle metabolism and fatigue during plyometric exercise and recovery. A review of the literature tends to show that studies either examine one of the aforementioned parameters individually or combine two of them for analysis. We hypothesized that short inter-set recovery may induce muscular fatigue. The ability to repeat high-intensity short-duration plyometric training following short recovery time is important for athletes.

## 2. Materials and Methods

### 2.1. Subjects

Ten male university basketball players, 20 ± 1 years of age, weighing 87.6 ± 7.6, and 190.3 ± 6.3 cm tall, were recruited in this study (Table 1). The sample size was determined by the power analysis. The parameters of power analysis were determined by previous studies [45]. The outcome of the power analysis was 8. We recruited 10 subjects in case of invalid data. The procedures undertaken in the study were approved by the University Ethics Committee and in accordance with the ethical standards specified by the Helsinki Declaration of 1975. All subjects signed consent forms to participate in the study after being informed about the procedures as well as possible risks involved in the whole project.

### 2.2. Procedures

Before data collection, the subjects completed three sessions of loaded CMJ plyometric exercise using an Olympic-sized bar with Olympic free weights on a fixed vertical plane and a smith-machine (Smith-machine, Beistegui Hermanos S.A., Set 0.04, Vitoria-Gasteiz, Spain). The smith-machine restricts the subject’s movement in the vertical plane; thereby assuring balanced movement and safety. Subjects practiced the loaded CMJ and demonstrated that they could perform the plyometric maneuvers correctly (Figure 1).

The subjects were instructed in the test procedures before the measurements were taken. In addition, several warm-up sets were performed prior to the actual tests. The value of 1 RM (1 repetition maximum) was defined as the subject’s maximum dynamic strength that can be performed by the subject. For the 1 RM test, subjects performed a counter-movement squat assuming an extended position (knee angle about 180°), while their shoulders were in contact with a bar. The subjects performed a controlled eccentric leg flexion to 90°, and from that position performed a concentric leg extension (as fast as possible) to reach full extension of 180° against the resistance determined by the weight plates added to both ends of the bar. The 1 RM CMJ squat test was used to determine lower body strength and to calculate the 1 RM loads used for exercise training. The correct CMJ jump requires a knee angle around 90° at the end of the counter movement. Trials were not considered valid if the knees didn’t bend to maximum intensity quickly without a break. The course of movement was inverted through the extension of the knees.

Previous studies suggested lower loads (<40 1 RM) and 6–10 repetitions per set induce greater improvements for athletes. Therefore, the plyometric exercises were individualized for each subject based on their 1 RM, the added weight was set at 30% 1 RM load in the present study. The subjects performed a controlled eccentric leg flexion to 90°, and from that position performed a concentric jump (as fast as possible and as high as possible) against the resistance determined by the weight plates added to both ends of the bar. Subjects performed 4 groups of exercise, each group consisting of 3 sets of 10 consecutive CMJ jumps with a 30% 1 RM load. Subjects were required to complete one set within 15 s without compromising proper technique. Each of the four 3-set groups had its own fixed inter-set recovery time, one with 1 min, one with 2 min, one with 3 min, and one with 5 min, during which time the subjects was standing still comfortably. The order of groups was randomized. Plenty of rest time (about 10 min) was given to the subjects between groups to avoid muscle fatigue.

### 2.3. Measurement

#### 2.3.1. Mechanical Data Collection

Testing the explosive power of lower limb muscles is a common practice in the assessment of human exercise performance. The mechanical data were collected from a force plate (Kistler, type 9286AA, Winterthur, Switzerland) with a sampling rate of 500 Hz. The key output data are the ground reaction forces and moments.

#### 2.3.2. Surface EMG Data Collection

Surface EMG (sEMG) activity of right lower limb muscles was recorded from the muscle bellies of *gluteus maximus* (GM), *rectus femoris* (RF), *vastus medialis* (VM), *vastus lateralis* (VL), *biceps femoris* (BF), *tibialis anterior* (TA), *lateral gastrocnemius* (LG), *medial gastrocnemius* (MG), and soleus (SO) using parallel-bar sEMG sensors (Trigno standard EMG sensor, Delsys Inc., Boston, MA, USA) at 2000 Hz, after removal of hair and cleaning with alcohol wipes. The placement of EMG electrodes followed the procedures recommended by the SENIAM Project [46]. The electrode application was based on lower limb dominant bone areas and prominences or other structures that can easily be palpated. We identified these points to locate the position of a selected electrode site. The 9 muscles are the major muscles activated during the CMJ jump. Fatigue levels were assessed for each muscle individually since the response to fatigue might be different during the CMJ jump. The range of knee angle was measured by a goniometer (SG 150, Biometrics Ltd., Ladysmith, VA, USA) (Figure 2). A goniometer adaptor (Trigno, Delsys, Inc., Natick, MA, USA) was used to synchronize goniometer data with EMG data.

#### 2.3.3. Near-Infrared Spectroscopy (NIRS) Data Collection

The portable NIRS apparatus (Portmon, Artinis Medical Systems, Amsterdam, The Netherlands) was attached to the motor point of the right vastus lateralis muscle along the vertical axis of the thigh, approximately 10 cm from the knee joint. The probe was wrapped in a piece of clear plastic foil to prevent the accumulation of sweat on its diodes. Then the NIRS and sEMG sensors were secured by wrapping a black elastic bandage around the thigh without occluding the blood flow.

The NIRS signals were sampled at 10 Hz, while the sEMG signals were sampled at 2000 Hz. A digital trigger was used to synchronize the recording of NIRS and sEMG signals.

### 2.4. Data Analysis

Exercise-related neuromuscular fatigue is defined as an exercise-related decrease in the maximal voluntary force or power of a muscle or a muscle group. It has been shown that decreases in movement peak explosive power during exercise are a valid indicator of neuromuscular fatigue [47].

The explosive power was calculated by jump height [48] and power [49], as determined from time in the air (*t*)
Jump height = 1/2 g (*t*/2)^2^(1)
where g = 9.81 m/sec^2^, *t* = time in air. Time in the air was determined from the ground reaction force data generated by the force plate.
Peak power = 51.9 × jump height + 48.9 × body mass (with added loads) − 2007 (2)

EMG signals were processed by wavelet analysis, which has proven to be a sensitive method of assessing non-stationary EMG signals. This method has been described in detail in previous papers [50]. Briefly, this method uses a filter bank of 10 non-linearly scaled wavelets, index by *k*, with center frequency, fc, ranging from 7 Hz (wavelet 0) to 395 Hz (wavelet 9). The first wavelet of EMG (*k* = 0) covered a frequency band of 0–10 Hz, which is typically associated with movement artifacts. So the first wavelet from spectra was excluded to reduce the effects of movement. 1 RM test was used to normalize the spectra for each subject. The frequency range, 10–350 Hz, covers nearly 95% of the power spectrum of EMG. The intensities from the wavelets (*k* = 1–9) were summed to give the EMG total intensity for each jump, each jump starting from an upright position. Total intensity is a measure of the time-varying power within the signal and is equivalent to twice the square of the root-mean-square. The instantaneous mean power frequency (MPF) was calculated by:(3)$$\mathrm{MPF}=\frac{\sum_{k}f_{c}(k)i_{k}}{\sum_{k}i_{k}}$$
with *f_c_* representing the center frequency of each wavelet, *i_k_* (*k* = 1–9) is the intensity of each wavelet. The mean frequency was calculated as the mean of the MPF values taken from each jump. Since the first and last jump was influenced by jump acceleration and deceleration, only jumps 2–8 were used for further calculation.

Changes in tissue oxy-hemoglobin ([HbO_2_]) and deoxy-hemoglobin ([HHb]) were measured using the differences in absorption characteristics of light at 750 and 850 nm by the portable NIRS apparatus. Changes in [HbO_2_] and [HHb] concentration were calculated by the modified Beer–Lambert law [51]. Values for [HbO_2_] and [HHb] measured are reported in micromolar units (μM). A moving average method (3 s) was used to smooth the NIRS signal. Tissue saturation index (TSI) was used to assess muscle (re)oxygenation level during exercise and inter-set recovery [42].
TSI = ([HbO_2_])/([HbO_2_] + [HHb]) × 100%(4)

TSI reflects the dynamic balance between O_2_ supply and O_2_ consumption and is independent of near-infrared photon path-length in muscle tissue. We decided to restrict our analyses to TSI rather than [HHb] in the present study because that provides a better indication of muscle oxygenation status when blood flow is not constant [52].

### 2.5. Statistics

Statistical analysis was performed using IBM SPSS Statistics (version 22, IBM, Inc., New York, NY, USA). Shapiro–Wilk test was used to assess data normality. One-way ANOVA with repeated measures was performed to compare EMG MPF, NIRS TSI values, and mechanical parameters across 3 sets. One-way ANOVA was then conducted to compare mean set values of explosive height, explosive power, EMG intensity, MPF, NIRS HbO_2_, and TSI among 4 groups. Bonferroni post-hoc multiple comparisons were then performed to assess significant differences determined by repeated-measures ANOVA. All data are reported in the text as mean (standard deviation (SD)). A significant level was set at *p* < 0.05 for all statistical procedures.

## 3. Results

The key kinetic data reported are peak explosive power and jump height. Table 2 shows the mean values of peak explosive power and jump height of each set for the 4 groups. If the lower extremity muscles become fatigued during the plyometric exercises, the peak explosive power should be decreased among the 3 inter-sets. However, peak explosive power and jump height were consistent among the 3 inter-sets and among the 4 groups. There were no significant differences in peak explosive power and jump height among sets and among groups (peak explosive power, *p* = 0.35; jump height, *p* = 0.17).

EMG indices, EMG intensity and MPF are used as indicators of muscle fatigue. Table 3 lists the EMG intensity of the 9 tested muscles at each set and group. When muscles are fatigued, the EMG intensity should increase whereas the MPF should decrease. However, we did not find significant differences in EMG intensity for the tested muscles among sets and among groups for GM (*p* = 0.34), RF *(p* = 0.57), VM *(p* = 0.43) VL (*p* = 0.55), BF (*p* = 0.09), TA (*p* = 0.56), MG (*p* = 0.82), LG (*p* = 0.84), and SO (*p* = 0.44). Table 4 shows the EMG mean power frequency (MPF) of the 9 tested muscles for each set and group. No significant differences in EMG MPF were found among sets and among groups for GM (*p* = 0.95), RF *(p* = 0.20), VM *(p* = 0.98) VL (*p* = 0.73), BF (*p* = 0.07), TA (*p* = 0.12), MG (*p* = 0.93), LG (*p* = 0.34), and SO (*p* = 0.46).

NIRS is a non-invasive method for measuring oxygenation concentration in the blood. Figure 2 displays an example of HbO_2_ variations during 10 jumps and 1 min recovery in one set. Before the start of exercises, the average of HbO_2_ values during 30 s was obtained as a baseline. The HbO_2_ decreased during 10 jumps (about 15 s) and recovered to the baseline in 1 min rest (Figure 2). The rates of HbO_2_ changes between baseline and recovery were not significantly different between 3 inter-sets and groups (*p* = 0.86).

Figure 3 shows the changes of the tissue saturation index (TSI) during plyometric exercise and various inter-set recovery periods. After the onset of the CMJ jumps, TSI decreases rapidly because of rapid usage of oxygen, and then TSI rises exponentially toward the baseline level. There are no significant differences in TSI at the end of the recovery periods, regardless of their 1 min, 2 min, 3 min, and 5 min duration (*p* = 0.21).

## 4. Discussion

The purpose of the study was to assess muscular activity with portable and non-invasive methods during plyometric training to avoid exercise-induced muscular fatigue. The recovery time between the groups of the plyometric training activities is usually 2–4 min [23,25,43,44], we hypothesized that a 1-min interval might not sufficient to allow the athletes to recover without inducing fatigue. Our hypothesis was not supported. One-minute inter-set recovery time did not reduce explosive power and did not induce muscle fatigue between 10 repetitions in one set. Muscle oxygenation could recover to baseline in approximately 1 min, similar to that of recovery time of 2 min, 3 min, and 5 min.

Decreases in maximal force and rate of force development are explained by muscular fatigue [53]. During heavy exercise, depending on the intensity, duration, and type of exercise, different amounts of force loss can occur. In the event of muscle fatigue, the muscle explosive power recovery might take hours or days. In contrast, if there is no muscle fatigue, peak explosive power recovers after inter-set recovery periods. Explosive power loss especially after eccentric exercise is one of the most valid and reliable indirect measures of muscle fatigue in humans [54]. We had expected that the explosive power and jump height would decrease because of muscle fatigue during plyometric exercises. In the present study, there were no significant differences in explosive power and jump height among sets and groups. The subjects maintained a rapid power development and muscle activation during all sets and groups. The 1 min recovery period between sets did not place subjects in a pre-fatigue state before the next set, as attested by the similar results of explosive power and jump height.

Surface EMG (sEMG) has been used to monitor muscular fatigue during exercises because of its non-invasive nature [28,55]. Changes in the EMG signals associated with fatigue are mean power frequency (MPF) and EMG amplitude [28]. Previous studies have shown that muscle fatigue is reflected in the EMG signal as an increase of its amplitude and a decrease of its characteristic spectral frequencies [56]. Although the mechanisms are not fully understood, some researchers have suggested that increased EMG intensity caused by muscle fatigue may reflect higher muscle fiber recruitment for a fixed external force [31,57]. The decreases in EMG MPF related to muscle fatigue are due to a shift towards lower frequencies, a small increase in low-frequency signal power, a relative decrease in high-frequency signal power, a decrease in low-frequency spectrum slope, or an increase in high-frequency spectrum slope [58]. In the present study, we did not find significant changes in EMG intensity and MPF among 3 sets and among 4 groups in any of the tested muscles. These results indicate that, during exercise sets and recovery periods, similar levels of muscle activities of the tested 9 muscles were observed for the 4 groups of plyometric training. Muscle fatigue of the tested 9 muscles was not detected by the sEMG signals in the present study.

Plyometric exercises rely mainly on anaerobic energy metabolism and more particularly on the phosphagen system (ATP and PCr) [59]. During recovery from exercise, oxygen consumption remains elevated, allowing the intramuscular high energy phosphates required to perform high-intensity exercise to be restored to pre-exercise levels [60,61]. Decreased oxygen availability can be expected to be a limiting factor of ATP and phosphocreatine (PCr) resynthesizing during recovery from exercises [62,63,64]. Newcomer et al. have reported reductions in muscular oxidative capacity after bouts of acute unaccustomed exercise, accompanied by increased rates of anaerobic ATP production, fatigue, and delayed onset muscle soreness [65]. Jones and Poole also observed that oxygen delivery status (such as blood flow) and PCr depletion kinetics could affect the kinetics of the initial muscle oxygenation at the onset of exercise [66]. This indicates the importance of oxygen availability and utilization in exercising muscles for recovery from exercise. Previous studies have suggested that oxygen stored in myoglobin has the time to be reloaded before the following exercise begins for high-intensity intermittent exercises [67]. As expected, plyometric training (10 CMJ jumps) in the present study induced a substantial muscle oxygen desaturation as measured by NIRS on vastus lateralis (Figure 2 and Figure 3), which is consistent between sets. Thereafter, the recovery of muscle oxygenation was almost complete in 1 min. The baseline values, the lowest values after 10 CMJ jumps for each set, the decline in the HbO_2_ and TSI, and the re-oxygenation during recovery periods were not significantly different between sets and groups (Figure 2 and Figure 3). These results indicate that, during exercise sets and recovery periods, similar levels of deoxygenation and re-oxygenation were observed for the 4 groups of plyometric training. The recovery time of muscle re-oxygenation may be associated with exercise mode and intensity. Previous studies reported that the recovery time is about 2 min after maximally 15-s running [42] or 30-s cycling sprints [38].

### Limitations

The current study has shown no muscular fatigue induced at relatively lighter loads CMJ Plyometric training. However, the findings of this study may be limited by a small sample size. Although the sample size might be small, we used kinetic values (explosive power, jump height), neuromuscular activity (EMG), and tissue oxygenation (NIRS) to detect the effects of inter-set recovery time on muscle fatigue during training. Previous studies either examine one of the aforementioned parameters individually or combine two of them for analysis. None of the parameters detected in the present study show significant results that suggesting muscle fatigue.

Training programs with low resistance are as effective for improving performance as heavy resistance training [68,69]. It has been proposed that plyometric exercise for added 60–90% 1 RM is too challenging and risky. In the present study, 30% 1 RM was assessed; however, fatigue is likely to occur at heavier loads. Future studies should investigate shorter inter-set recovery times and different loads to shed light on the influence of both rest period frequency and duration on neuromuscular fatigue during plyometric training.

We adopted the training methods which have been used by a local University basketball team. They normally undertake 3 sets of 10 CMJ jumps with 30% 1 RM loads. However, our results may not be extrapolated to other training methods, regarding different jump types, repetitions, sets, and groups. Additional studies are required to investigate the relative importance of muscle activity and tissue oxygenation in different protocols and their long-term adaptation in response to training.

## 5. Practical Applications

During the short-duration CMJ explosive training, a 1 min inter-set recovery time could be adopted because no muscle fatigue was observed in terms of EMG activity and muscle blood oxygenation, and the peak power output was well maintained.When performing smith squat CMJ jumps at the loads at 30% 1 RM, each training could consist of 3 sets, and 10 repetitions can be performed without inducing muscle fatigue and a significant decrease in CMJ velocity and peak power output.

## 6. Conclusions

Application of kinetics, sEMG, and NIRS as three integrated assessing tools could conveniently disclose many mechanical and physiologic phenomena of human muscle during plyometric training for athletes. The recommended inter-set recovery periods for plyometric training are between 2 min and 4 min. Our results suggested that a 1 min inter-set recovery period is enough for a set with 10 repetitive CMJ jumps with 30% 1 RM loading. Information about muscle activity and recovery would help coaches and players design training protocols. Considering that the training program applied in this study was of short duration (about 5 min for 3 sets) and produced low fatigue levels, coaches and players could include a short training session as their daily training routine to enhance strength and explosive power.

## Figures and Tables

**Figure 1 sensors-21-03015-f001:**
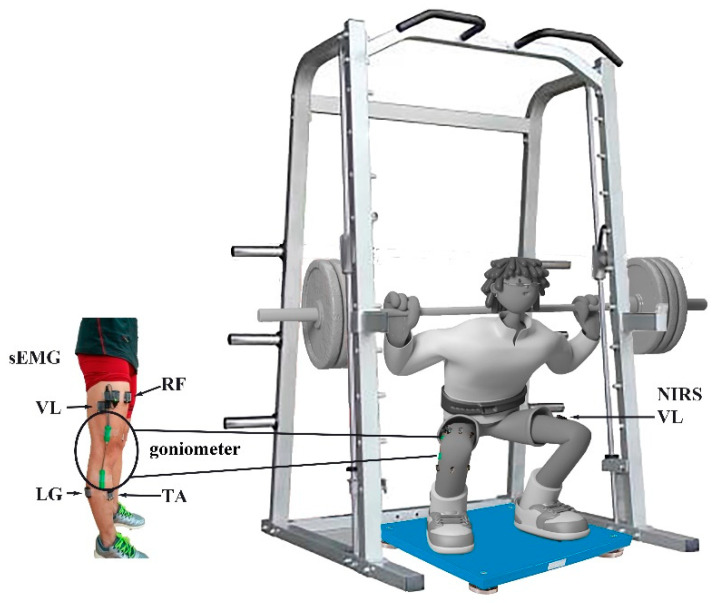
Experimental setup and the instrumentation used during the 30% 1 RM loaded CMJ jump. sEMG sensors were attached to 9 lower muscles and an SG 150 goniometer was placed on the right side; a NIRS sensor was attached to the VL on the left side, and a force plate was placed under the athlete’s feet.

**Figure 2 sensors-21-03015-f002:**
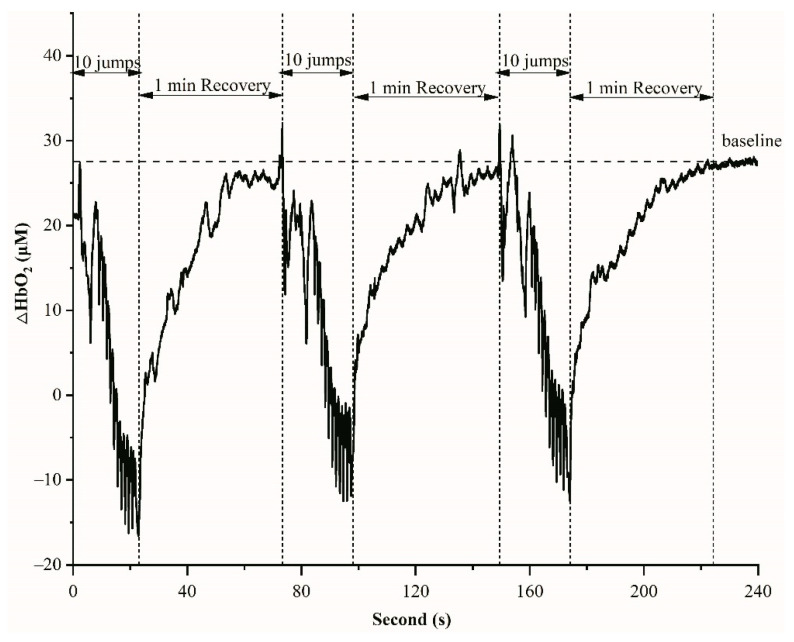
An example of HbO_2_ variations during 10 jumps and 1 min recovery in one set.

**Figure 3 sensors-21-03015-f003:**
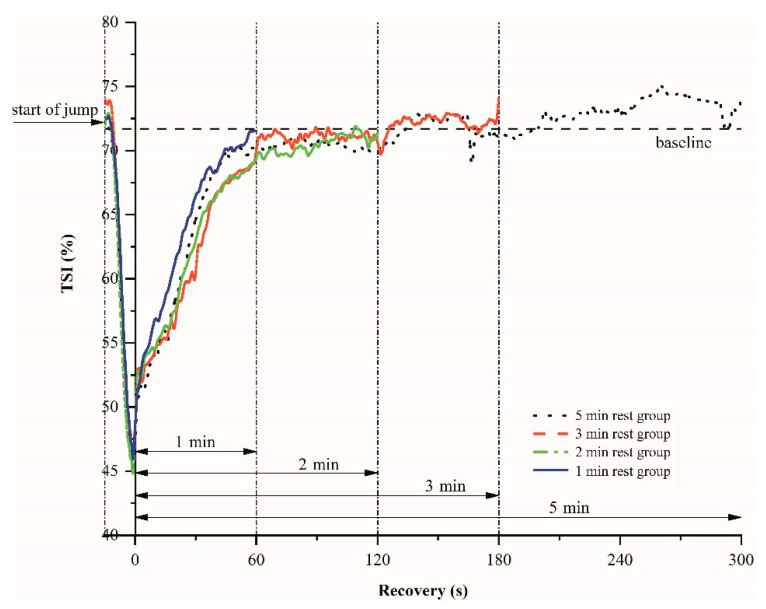
An example of the tissue saturation index (TSI) variations-time relations during plyometric exercise and various recovery periods.

**Table 1 sensors-21-03015-t001:** Physical and physiological characteristics of subjects.

Subject Code	Age (Year)	Height (cm)	Weight (kg)	1 RM (kg)	Training Experience (Year)
A	22	180	77	130	6
B	21	188	90	120	5
C	20	186	90	110	4
D	21	198	97	130	5
E	21	182	85	100	5
F	22	195	80	140	6
G	20	189	101	140	4
H	20	196	86	120	4
I	21	196	90	100	5
J	19	193	80	100	4
Mean ± SD	20 ± 1	190.3 ± 6.3	87.6 ± 7.6	119.0 ± 15.1	4.8 ± 0.7

Note: 1 RM: 1 repetition maximum.

**Table 2 sensors-21-03015-t002:** Mean set values of the peak explosive power and jump height for the 4 groups.

		Peak Explosive Power (W)	Jump Height (cm)
5 min	Set 1	2549.2 ± 463.0	19.57 ± 4.27
	Set 2	2522.6 ± 445.1	20.06 ± 4.12
	Set 3	2522.7 ± 445.8	20.09 ± 3.33
3 min	Set 1	2522.0 ± 445.3	18.90 ± 3.77
	Set 2	2521.8 ± 445.1	18.70 ± 2.96
	Set 3	2454.3 ± 386.3	19.00 ± 3.41
2 min	Set 1	2521.6 ± 445.1	18.25 ± 2.37
	Set 2	2521.7 ± 444.9	18.47 ± 1.75
	Set 3	2521.5 ± 444.8	18.06 ± 1.89
1 min	Set 1	2521.9 ± 445.1	18.86 ± 2.62
	Set 2	2521.9 ± 445.5	18.83 ± 2.27
	Set 3	2522.3 ± 445.6	19.55 ± 1.98

**Table 3 sensors-21-03015-t003:** Normalized EMG intensity for the 9 tested muscles for each set and group.

Normalized Intensity (%)		GM	RF	VM	VL	BF	TA	MG	LG	SO
5 min	Set 1	41.2 ± 6.5	45.7 ± 6.5	48.6 ± 4.6	47.8 ± 6.9	35.5 ± 4.4	43.4 ± 5.6	51.9 ± 6.1	47.0 ± 7.0	53.8 ± 3.2
	Set 2	38.4 ± 6.6	38.9 ± 5.8	39.4 ± 2.7	41.9 ± 5.1	37.0 ± 5.3	33.1 ± 4.4	43.7 ± 4.0	48.2 ± 7.6	45.2 ± 4.6
	Set 3	34.2 ± 5.9	37.3 ± 6.2	38.9 ± 4.6	40.3 ± 6.1	45.0 ± 7.2	36.7 ± 5.5	39.4 ± 7.1	41.1 ± 5.5	33.9 ± 4.3
3 min	Set 1	46.0 ± 6.3	46.3 ± 6.4	51.2 ± 4.9	49.2 ± 4.5	36.7 ± 5.8	41.2 ± 5.5	39.6 ± 5.9	45.4 ± 4.1	43.7 ± 5.0
	Set 2	41.9 ± 5.6	40.0 ± 5.1	43.0 ± 3.6	46.3 ± 3.7	38.5 ± 4.1	47.5 ± 3.6	49.7 ± 7.2	44.6 ± 4.3	35.9 ± 4.3
	Set 3	37.1 ± 3.7	43.9 ± 4.2	35.0 ± 3.7	44.3 ± 5.0	37.7 ± 3.0	37.8 ± 4.2	46.3 ± 4.8	44.2 ± 5.8	39.8 ± 2.7
2 min	Set 1	48.2 ± 4.9	56.2 ± 4.4	49.8 ± 6.3	46.7 ± 5.4	38.3 ± 4.9	39.9 ± 4.5	38.3 ± 5.4	48.7 ± 4.4	47.3 ± 3.8
	Set 2	43.7 ± 4.2	42.9 ± 3.5	37.8 ± 4.8	36.4 ± 3.5	36.8 ± 5.0	40.1 ± 4.5	46.5 ± 3.8	47.1 ± 3.8	51.1 ± 6.9
	Set 3	39.5 ± 2.0	43.4 ± 2.9	39.5 ± 4.9	42.3 ± 5.3	36.3 ± 4.1	35.6 ± 4.0	43.2 ± 4.9	37.6 ± 5.3	43.7 ± 5.3
1 min	Set 1	37.9 ± 2.8	38.2 ± 5.5	46.6 ± 6.3	39.6 ± 4.9	42.5 ± 6.3	37.5 ± 2.2	45.5 ± 3.2	45.7 ± 5.0	43.2 ± 0.9
	Set 2	42.8 ± 3.9	42.1 ± 3.2	52.1 ± 4.9	42.6 ± 4.7	42.1 ± 5.0	41.0 ± 3.0	42.3 ± 6.5	48.5 ± 5.6	37.2 ± 4.0
	Set 3	39.2 ± 3.8	51.6 ± 4.2	50.1 ± 5.0	47.8 ± 5.8	45.1 ± 4.3	43.1 ± 5.0	52.5 ± 5.5	46.7 ± 4.6	42.4 ± 3.9

Table note: GM: gluteus maximus; RF: rectus femoris; VM: vastus medialis; VL: vastus lateralis; BF: biceps femoris; TA: tibialis anterior; MG: medial gastrocnemius; LG: lateral gastrocnemius; SO: soleus.

**Table 4 sensors-21-03015-t004:** EMG mean power frequency (MPF) of the 9 tested muscles at each set and group.

MPF(Hz)		GM	RF	VM	VL	BF	TA	MG	LG	SO
5 min	Set 1	43.0 ± 5.3	43.9 ± 5.3	39.4 ± 3.6	47.1 ± 6.7	48.1 ± 5.8	50.0 ± 5.7	51.4 ± 6.8	50.4 ± 5.1	45.6 ± 2.8
	Set 2	46.2 ± 3.0	50.5 ± 4.0	55.0 ± 4.2	53.5 ± 5.0	45.0 ± 2.8	50.4 ± 5.7	52.3 ± 4.3	53.7 ± 3.8	47.8 ± 4.0
	Set 3	57.6 ± 5.3	54.7 ± 7.8	54.5 ± 3.0	52.5 ± 6.1	44.1 ± 7.1	50.9 ± 5.6	50.2 ± 5.6	57.2 ± 4.1	53.3 ± 5.6
3 min	Set 1	54.0 ± 6.3	53.3 ± 4.9	38.5 ± 4.7	40.5 ± 2.9	51.0 ± 3.1	64.7 ± 6.8	54.7 ± 5.7	59.1 ± 6.7	49.8 ± 5.6
	Set 2	49.1 ± 4.1	56.5 ± 4.7	48.6 ± 3.1	49.2 ± 3.1	47.3 ± 3.7	63.3 ± 5.1	44.2 ± 5.2	48.6 ± 8.9	51.2 ± 5.4
	Set 3	43.0 ± 5.7	52.6 ± 4.5	58.6 ± 3.7	51.8 ± 4.2	52.2 ± 4.2	53.0 ± 5.2	50.6 ± 4.2	50.6 ± 5.6	45.3 ± 4.7
2 min	Set 1	50.3 ± 3.2	54.5 ± 6.0	43.2 ± 5.2	44.8 ± 2.7	54.1 ± 5.7	57.8 ±4.3	51.8 ± 4.1	50.7 ± 5.0	54.9 ± 3.0
	Set 2	49.5 ± 4.0	44.2 ± 5.1	53.2 ± 4.0	52.1 ± 2.7	52.9 ± 3.2	51.5 ± 3.0	53.4 ± 3.1	49.9 ± 5.7	52.6 ± 2.4
	Set 3	47.0 ± 4.1	49.9 ± 2.2	52.4 ± 2.8	47.4 ± 4.6	49.9 ± 4.3	44.4 ± 4.1	49.3 ± 3.1	51.4 ± 4.9	53.0 ± 4.7
1 min	Set 1	53.4 ± 4.0	48.9 ± 5.1	41.7 ± 5.9	52.3 ± 3.1	50.9 ± 5.0	56.6 ± 5.0	55.1 ± 4.3	51.6 ± 4.2	59.1 ± 3.5
	Set 2	48.1 ± 4.7	46.7 ± 3.2	45.3 ± 3.0	47.6 ± 4.2	45.6 ± 3.1	53.6 ± 3.4	48.0 ± 4.3	45.6 ± 4.5	50.2 ±2.2
	Set 3	51.0 ± 3.3	39.3 ± 5.4	54.9 ± 5.9	42.6 ± 4.0	42.1 ± 4.0	52.1 ± 4.1	50.4 ± 4.9	48.1 ± 4.2	46.6 ± 3.7

Table note: GM: gluteus maximus; RF: rectus femoris; VM: vastus medialis; VL: vastus lateralis; BF: biceps femoris; TA: tibialis anterior; MG: medial gastrocnemius; LG: lateral gastrocnemius; SO: soleus.

## Data Availability

Please contact the corresponding author if raw data is required.

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
