# Peer review of "The Effects of Inter-Set Recovery Time on Explosive Power, Electromyography Activity, and Tissue Oxygenation during Plyometric Training"

_sensors, 2021, doi:10.3390/s21093015_

Round 1

Reviewer 1 Report

The purpose of this study was to investigate the effects of inter-set recovery time on lower extremity explosive power, neuromuscular activity, and tissue oxygenation during plyometric exercise and recovery. An integrated data collection and analysis of explosive power, electromyography activity, and tissue oxygenation is necessary for understanding local muscle metabolism and fatigue during plyometric exercise and recovery. A review of the literature tends to show that studies either examine one of the aforementioned parameters individually or combine two of them for analysis.

Application of kinetics, sEMG and NIRS as three integrated assessing tools could conveniently disclose many of mechanical and physiologic phenomenon of human muscle.

Reviewer's comments:

1) How was the range of the knee angle tested, was it a subjective assessment or were objective measurement tools used?

2) I understand that all participants were right-footed. It would be interesting to study the muscle fatigue reactions in both lower limbs.

3) Line 189 – „The first wavelet of EMG (k = 0) covered a frequency band of 0-10 Hz, which is typically associated with movement artefacts” - What basis was such a statement made? Is it supported by the latest reports?

Reviewer 2 Report

Authors intended to investigate the effects of inter-set recovery time on physiological parameters of lower limbs (EMG, power and oxygenation) when performing plyometric training. The experimental protocol involved ten young healthy basketball players and no significative differences were found among the four tested exercise group. Topic is interesting, also considering the amount of already published papers.

Due to the lack of evidence, it is difficult to understand the real scientific impact of this paper with respect the published ones. In fact, the main outcome is “The results of this study may help inform recommendations concerning the recovery time during 23 plyometric exercises”; however such evaluation has been already performed for example in the cited papers 23, 25, 43 and 44. Authors must necessarily make clear the scientific contribution of the present paper with respect the literature.

Other issues:

  • Lines 109-111: Please report the outcome of the power analysis for the selection of the size group.
  • Please report the compliance with the Helsinki Declaration.
  • Is there any reference or justification for the selection of 30% of 1RM load?
  • Are authors sure that the randomization of exercise group did not lead to bias on the results? May players who rested more time in the first groups incur in lower fatigue? Please try to explain this.
  • Please with consistence with the GUM for the unit of measure, a space is generally required between the value and the measure unit.
  • More details on the EMG placement should be added. Did authors follow the SENIAM guidelines?
  • A figure with the experimental setup could allow readers to better understand the used instrumentation.
  • Please correct line 216.

Round 2

Reviewer 2 Report

Authors answered to my previous concerns